# Oregano Essential Oil as a Natural Plant Additive Affects Growth Performance and Serum Antibody Levels by Regulating the Rumen Microbiota of Calves

**DOI:** 10.3390/ani14060820

**Published:** 2024-03-07

**Authors:** Zhihao Luo, Ting Liu, Dongzhu Cairang, Shuru Cheng, Jiang Hu, Bingang Shi, Hui Zhu, Huan Chen, Tao Zhang, Xuejiao Yi

**Affiliations:** 1College of Animal Science and Technology, Gansu Agricultural University, Lanzhou 730070, China; lzhxxx0104@163.com (Z.L.); crdz2384580125@163.com (D.C.); chengsr@gsau.edu.cn (S.C.); huj@gsau.edu.cn (J.H.); shibg@gsau.edu.cn (B.S.); 17358111180@163.com (H.Z.); 17723395157@139.com (H.C.); ltsecjpsktgu7@163.com (T.Z.); yxj18794832346@163.com (X.Y.); 2College of Veterinary Medicine, Gansu Agricultural University, Lanzhou 730070, China; 3Gansu Key Laboratory of Herbivorous Animal Biotechnology, Lanzhou 730070, China

**Keywords:** calf health, oregano essential oil, immunoglobulin, rumen microorganisms

## Abstract

**Simple Summary:**

The health status and survival of calves are essential factors affecting the farm’s production efficiency and economic efficiency; the main reasons threatening the health of calves are the slow development of the gastrointestinal tract and the low immunity of ruminants at a young age. The coordinated action of the host and its gastrointestinal microflora can improve these negative factors. To better enable the gastrointestinal microflora to play a positive role in the animal organism, the composition and abundance of microorganisms can be regulated in vitro by supplementation with exogenous substances. Oregano essential oil, a natural plant extract, has been reported to be a suitable growth promoter for livestock and poultry, which can promote growth and development, enhance gastrointestinal function, and improve immunity by regulating the gastrointestinal microbiota of ruminants. We investigated oregano essential oil as an exogenous supplement for regulating rumen microbiology in calves and found that it promotes VFA production and can influence calf growth performance and serum antibody levels.

**Abstract:**

This experiment aimed to investigate whether supplementation of calves with different doses of oregano essential oil (OEO) could promote the development of the gastrointestinal tract and enhance the immune ability of calves by regulating the rumen microbiota. Twenty-four 70-day-old healthy and disease-free Holstein male calves were randomly divided into four groups, with the control group fed a basal diet, and the treatment group provided 4 g, 6 g, and 8 g of oregano essential oil per day in addition to the basal diet. After the 14-day pre-test, a 56-day formal test was conducted. At days 0 and 56 of the standard test period, calves were weighed, the average daily weight gain of calves during the test period was calculated, and serum samples were collected to measure the concentration of immunoglobulins (IgA, IgG, and IgM) in the serum; at day 56 of the formal test period, rumen fluid was collected from the calves, and 16SrRNA was sequenced to analyze changes in the rumen microbiota of the calves. The changes in the rumen microbiota of calves were analyzed by 16SrRNA sequencing. The results of the study showed that (1) OEO supplementation in calves significantly increased end weight and average daily gain (*p* < 0.05); (2) OEO supplementation in calves significantly increased serum concentrations of immunoglobulins IgA and IgM (*p* < 0.05); (3) OEO supplementation in calves significantly increased the abundance and diversity of rumen microbial organisms (*p* < 0.05); (4) OEO supplementation in calves significantly regulates the relative abundance of some species, and biomarkers with significant differences were screened by LEfSe analysis: g_Turicibacter, *g_Romboutsia*, f_Peptostreptococcaceae, f_Clostridiaceae, *g_Clostridium_sensu_stricto_1*, o_Clostridiales, *g_unclassified_f_Synergistaceae*, c_Coriobacteriia, o_Coriobacteriales, f_Atopobiaceae, *g_Olsenella*, p_Actinobacteriota, *g_Defluviitaleaceae_UCG-011*, f_Defluviitaleaceae, o_Corynebacteriales, *g_Corynebacterium*, f_Corynebacteriaceae, *g_Shuttleworthia*, f_Hungateiclostridiaceae, o_norank_c_Clostridia, *g_Saccharofermentans*, *g_Streptococcus*, f_Streptococcaceae, *g_unclassified_o_Oscillospirales*, and f_unclassified_o_Oscillospirales (*p* < 0.05, LDA ≥ 3); and (5) OEO supplementation in calves significantly enriched the metabolism of cofactors and vitamins pathway (*p* < 0.05). (6) Using Superman’s correlation analysis, we screened *unclassified_c_Clostridia*, *Shuttleworthia*, and *Christensenellaceae_R-7_group,* three beneficial strains for calves. (7) Daily supplementation with 8g of OEO significantly affected rumen microbiota regulation in calves.

## 1. Introduction

Proper nutritional supply and management practices are critical for calves’ growth performance, immunocompetence, and health. Good health reduces calf morbidity and mortality. A significant threat to calf health is the slow development of the gastrointestinal tract at an early age, which prevents calves from obtaining sufficient nutrients from the feed to improve GI function and build up adequate immunity. One way to promote the development of the gastrointestinal tract in calves is to provide them with dietary supplements [1], which can influence the growth and development of the gastrointestinal tract by regulating the microbial community in the calf’s gastrointestinal tract [2].

The rumen is a digestive organ unique to ruminants and is the most potent natural fermenter known to date for the degradation of fibrous material. The rumen has a unique and complex structure and function. Its most essential functions for ruminants are converting feed into energy for the animal body and altering substances such as lignocellulose to volatile fatty acids and other end products [3]. These processes are mainly driven by the rumen microbiome and its interaction with the host [4]. The microbial composition of the gastrointestinal tract of young ruminants is closely related to their growth, development, and health status [5], and the microorganisms in the rumen of ruminants are dominated by bacteria, fungi, protozoa, and archaea, which, through interactions and coordination, provide nutrients and regulate the immune response of the organism [6]. For ruminants, the role of microorganisms is not only to improve development and immunocompetence at a young age but also to have a long-term effect on the animal organism [7].

Since the 1950s, antibiotics have been used as growth promoters in the world’s livestock industry [8]. Although antibiotics can enhance animal growth performance, prevent and treat livestock and poultry diseases, and bring higher productivity and economic benefits to the livestock industry, with the development of animal husbandry science, the drawbacks of using antibiotics have gradually been revealed [9]; under the conditions of intensive farming, antibiotics have been widely and heavily used, and the antibiotic residues that the animals are unable to digest and metabolize enter into water sources or the soil along with the excretions of the body, which poses health risks to the environment and human beings [10]. Animal feces are the primary source of antibiotics in the environment [10,11]. For the sustainable development of animal husbandry, the environment, and human beings, using growth-promoting drug feed additives (except for traditional Chinese medicines) in livestock breeding is banned worldwide. Therefore, researchers are currently targeting the development of a new green and harmless growth promoter that can replace antibiotics and at the same time achieve the purpose of improving animal feed intake, growth performance, promoting the establishment of gastrointestinal digestion and immune function, and obtaining high-quality and safe animal food.

Plant essential oils, probiotics, prebiotics, and acidifiers are the main alternatives to growth-promoting pharmaceutical feed additives in livestock and poultry farming [12]. Oregano is a medicinal, aromatic plant belonging to the genus Oregano in the family Labiatae, with its main distribution area located in the Mediterranean Sea [13]. It is widely used in spice making and winery for its ability to give off a distinctive and rich aroma [14]. Oregano is rich in volatile compounds such as carvacrol, thymol, and monoterpenes, which exhibit different pharmacological properties such as anti-bacterial, anti-fungal, anti-viral, and anti-oxidant; it promotes vasodilatation and is hepatoprotective [15], so oregano is also used in traditional medicine and Chinese medicine to treat digestive, neurological and respiratory disorders [16]. Clinical studies have demonstrated the pharmacological potential of oregano in various properties, such as anti-diabetic and hyperlipidemic, cardioprotective and vascular, anti-inflammatory, anti-bacterial, and insecticidal [17]. Oregano can extract essential oils rich in carvacrol and thymol, a powerful natural antibiotic [18]. The ratio of its primary action components, carvacrol and thymol, is also the leading indicator of the bactericidal effect of oregano essential oils, which are a safe, green, highly effective, and non-contraindicative natural additive in traditional Chinese medicine. Oregano essential oil has been reported to improve growth performance in ruminants [19], increase apparent digestibility [20], promote the construction of immune function [21], promote gastrointestinal tract development in young animals [22], improve the anti-oxidant capacity of the organism [23], act as an anticoccidial [24], and regulate the microecology of the gastrointestinal tract [25]. This experiment investigated whether oregano essential oil as a dietary supplement could promote calf growth and development and improve antibody levels by regulating the rumen microbiota.

## 2. Materials and Methods

### 2.1. Experimental Design and Feeding Program

The experiment was conducted from July to September 2022 at Liaoyuan Breeding Base (Linxia, China), where the average temperature was 24.5 °C and the average humidity was 49.0% RH during the experimental period. The test subjects were 24 Holstein male calves of 70 days of age. Individuals suffering from respiratory diseases (colds, pneumonia), digestive diseases (diarrhea, dyspepsia), and hoof diseases were pre-excluded, and all calves were randomly divided into four groups. The control group (CON) was fed with a basal diet, and the treatment group was supplemented with a daily supplement of 4 g of oregano essential oil (LOW), 6 g of oregano essential oil (MID), and 8 g of oregano essential oil (HIG) in addition to feeding the basal diet. Oregano essential oil was purchased from Ralco Nutrition (Marshall, MN, USA); the main active ingredients are carvacrol (51.78%) and thymol (6.93%), with calcium carbonate, silica carbonate, and diatomaceous earth as the carrier. At the end of the daily morning feed, oregano essential oil was gavaged to the calf.

A 14-day pre-test was first conducted to acclimatize calves to additive feeding. Calves were 84 days old at the end of the pre-test period, followed by a 56-day formal trial ending at 140 days old. During the trial, the calf was housed individually in a calf pen with free access to feed and water. All calves were fed colostrum twice at birth (3.5–4 L within 1 h after birth and a second feeding of 2 L 6 h after birth) and vaccinated according to the farm’s standard operating procedures. Early weaning was practiced at 56 days of age.

### 2.2. Nutritional Level of Feed

The calf basal feed consisted of pellets and oat grass formulated at 5:1; nutrient levels of the basal feed are shown in Table 1. The pellets were purchased from Dehua Bio Co., Ltd. (Lanzhou, China).

### 2.3. Sample Collection and Measurement

#### 2.3.1. Growth Performance Measurement

On days 0 and 56 of the trial period, the body weight of each test calf was determined in the morning before feeding using a floor scale, and the average daily gain (ADG) of the calves was calculated at the end of the trial.

#### 2.3.2. Immunoglobulin Assay

On day 56 of the trial period, 10 mL of blood samples were collected from each calf using the jugular vein blood collection method, left at room temperature for half an hour, centrifuged at 3000 rmp for 10 min, and then the serum was separated and stored at −20 °C. At the end of the test period, the samples were sent to Huaying Biologicals (Beijing, China) for the determination of the immunoglobulin (IgA, IgG, IgM) content in the serum.

#### 2.3.3. Sequencing of Rumen Microorganisms

On day 56 of the trial period, 20 mL of rumen fluid was collected from each calf using the ruminal probe sampling method, filtered through four layers of sterile gauze, stored in liquid nitrogen, and subjected to DNA extraction and sequencing at the end of the test, using the following procedures:

DNA was extracted using the E.Z.N.A.^®^ Soil DNA Kit DNA Extraction Kit (Omega Bio-tek, Norcross, GA, USA) according to the product instructions, and then the concentration and purity of DNA was checked by NanoDrop2000 (Thermo Scientific Inc., Waltham, MA, USA), and the integrity of DNA was checked by 1% agarose gel electrophoresis. This DNA was then used as a template for PCR amplification of the V3V4 variable region of the 16S rRNA gene, with upstream primer 338F (5′-ACTCCTACGGGGAGGCAGCAG-3′) and downstream primer 806R (5′-GGACTACHVGGGTWTCTAAT-3′). The PCR instrument used was the ABI GeneAmp^®^ Model 9700 (ABI, Los Angeles, CA, USA). The reaction system was as follows: 5 × TransStart FastPfu buffer 4 μL, 2.5 mM dNTPs 2 μL, upstream primer (5 μM) 0.8 μL, downstream primer (5 μM) 0.8 μL, TransStart FastPfu DNA polymerase 0.4 μL, BSA 0.2 μL, and template DNA 10 ng, making up 20 μL. All samples were amplified in triplicate. The PCR product was extracted from 2% agarose gel and purified using the AxyPrep DNA Gel Extraction Kit (Axygen Biosciences, Union City, CA, USA) according to manufacturer’s instructions before being quantified using a Quantus™ Fluorometer (Promega, Madison, WI, USA). The purified PCR products were library-constructed using a NEXTFLEX Rapid DNA-Seq Kit (PerkinElmer, Waltham, MA, USA) and sequenced on the Miseq PE300 platform.

### 2.4. Statistics and Analysis of Data

The experimental data were statistically analyzed using SPSS 25.0 software, a one-way ANOVA was performed, and the *p* < 0.05 value was used as the significance marker. High-throughput sequencing data were quality-controlled using fastp 0.19.6 for double-ended raw sequencing sequences and spliced using FLASH 1.2.11 software, and optimized sequences after quality-control splicing were spliced using the Qiime2 process. The optimized sequences after QC splicing were subjected to noise reduction using the DADA2 plug-in, which removes the annotations to chloroplast and mitochondrial sequences from all samples and flattens the number of sequences in all samples according to the minimum number of sequences in the sample. After the flattening, the average sequence coverage (Good’s coverage) of each sample can still reach 99.66%. Based on the Sliva 16S rRNA gene database (v 138), OTUs were analyzed for species taxonomy using the naive Bayes classifier in Qiime2, the Alpha diversity index was calculated using mothur1.30.2, and the Wilcoxon rank sum test was used for the analysis of between-group differences in Alpha diversity. We used an algorithm based on the Bray–Curtis distance PCoA to analyze the similarity of microbial community structure between samples and combined it with a PERMANOVA non-parametric test to analyze whether the differences in microbial community structure between sample groups were significant. LEfSe analysis (LDA > 3, *p* < 0.05) was used to identify bacterial taxa with significant differences in abundance from phylum to genus level among different groups. PICRUSt2(2.2.0) was used to predict microbial functions, as well as a correlation analysis based on Spearman’s coefficient and construction of species correlation network diagrams (|r| > 0.5, *p* < 0.05); The results of the analyses of between-group diversity and between-group species difference were corrected for multiple testing using FDR.

## 3. Results

### 3.1. Effect of Oregano Essential Oil Supplementation on Calf Growth Performance

As can be seen in Table 2, there was no difference in calf weight between groups at the beginning of the trial (*p* > 0.05), and after supplementation with oregano essential oils, calf final weight and ADG were higher (*p* < 0.05) compared to CON. Between treatment groups, LOW and MID were lower (*p* < 0.05) than HIG. There was no difference between MID and CON and LOW (*p* > 0.05).

### 3.2. Effect of Oregano Essential Oil Supplementation on Calf Serum Immunoglobulin

As shown in Table 3, there was no difference in serum immunoglobulin concentration of calves in each group at the beginning of the experiment (*p* > 0.05). At 56 days, IgA concentration in HIG group was higher than that in CON (*p* < 0.05), and there was no difference among the remaining three groups (*p* > 0.05). IgG showed no difference among all the treatment groups (*p* > 0.05). The IgM concentration was higher in the HIG and MID groups than that in CON and LOW (*p* < 0.05), and there was no difference between the HIG and MID and CON and LOW groups (*p* > 0.05).

### 3.3. Calf Rumen Microbial Base Sequencing Results

A total of 1,578,347 and 650,336,397 bases of optimized sequences were obtained from sequencing, with an average sequence length of 412 bp. The sequences obtained from sequencing were drawn flat according to the minimum number of sample sequences and clustered according to the silva138/16s_bacteri species taxonomic database and the USEARCH11-sparse algorithm, with a classification confidence of 0.7 and the OTU sequence. The annotation results were as follows: Domain:1; Kingdom:1; Phylum:17; Class:30; Order:77; Family:138; Genus:273; Species:489; OTU:1907. When the species dilution curve (Figure 1A) flattens, it indicates that the amount of sequencing data is large enough to reflect the vast majority of microbial diversity information in the sample. According to the species Venn diagram (Figure 1B), there were 1042 OTUs and 150 endemic OTUs in CON, 1121 OTUs and 177 endemic OTUs in LOW, 1096 OTUs and 145 endemic OTUs in MID, and 1300 OTUs and 296 endemic OTUs in HIG.

### 3.4. Analysis of Rumen Microbial Diversity in Calves

#### 3.4.1. Alpha Diversity

There were differences (*p* < 0.05) in the species richness Chao1 index (Figure 2A) and diversity Shannon and Simpson indices (Figure 2B,C), and no differences (*p* > 0.05) in the spectral diversity PD index (Figure 2D) and species coverage index (Figure 2E). The calf rumen microorganism samples in the present experiment all had a coverage index higher than 0.99, indicating that the results of this sequencing could represent the actual situation of the flora in the samples (Figure 2E).

#### 3.4.2. Beta Diversity

Using PCoA principal coordinate analysis, the similarity of microbial community composition between samples was studied. The results showed that there was partial overlap between samples from different groups. Still, no significant separation was observed, with *p* = 0.407 and R = 0.0078, indicating that the differences in microbial communities between and within sample groups were relatively small (Figure 3).

### 3.5. Analysis of Microbial Species Composition in the Rumen of Calves

The species composition of microorganisms at the phylum and genus levels were similar in all groups (Figure 4A,B), and the top five species in terms of relative abundance at the phylum level were Firmicutes, Actinobacteriota, Bacteroidota, Patescibacteria, and Proteobacteria (Table 4). The relative abundance of Actinobacteriota was different among the groups (*p* < 0.05); the top ten species in terms of relative abundance at the genus level were *Olsenella*, *Erysipelotrichaceae_UCG-002*, *Acetitomaculum*, *norank_f__norank_o__Clostridia_UCG-014*, *Prevotella*, *Lachnospiraceae_NK3A20_group*, *Ruminococcus_gauvreauii_group*, *Shuttleworthia*, *Syntrophococcus*, *unclassified_f__Lachnospiraceae*. The relative abundance of *g_Olsenella* and *g_Shuttleworthia* was different between groups (*p* < 0.05).

### 3.6. Analysis of Differences in Microbial Species in the Rumen of Calves

Analyzing the difference in species at multiple levels using the LESfe method means we can screen different groups of calves for significantly different species. In this experiment, the differential species were screened by comparing the magnitude of the effect of rumen microbial species abundance with the differential effect in each group using the LDA effect value and *p*-value (LDA score ≥ 3, *p* < 0.05), and the results showed that (Figure 5A,B) the differential biomarker in the CON group was *Erysipelatoclostridiaceae*. In the LOW group, the differential biomarkers were *g _Turicibacter*, g_Romboutsia, f_Peptostreptococcaceae, f_Clostridiaceae, *g_Clostridium_sensu_stricto_1*, o_Clostridiales, and *g_unclassified_f_Synergistaceae*. The differential biomarkers in the MID group were c_Coriobacteriia, o_Coriobacteriales, f_Atopobiaceae, g_Olsenella, p_Actinobacteriota, *g_Defluviitaleaceae_UCG-011*, f_Defluviitaleaceae, o_Corynebacteriales, *g_Corynebacterium*, and f_Corynebacteriaceae. The HIG group differential biomarkers were *g_Shuttleworthia*, f_Hungateiclostridiaceae, o_norank_c_Clostridia, *g_Saccharofermentans*, *g_Streptococcus*, f_Streptococcaceae, *g_unclassified__o__ Oscillospirales*, and f_unclassified_o_Oscillospirales.

### 3.7. Prediction of Rumen Microbial Function in Calves

PICRUSt2 was used to predict the functional information of communities in calf rumen microorganisms, and the functional composition and abundance can be used to understand the potential microbial functional characteristics. The prediction results showed that the functional compositions of the groups were similar on KEEG Pathway Level 2, in which the top ten pathways with functional abundance were global and overview maps, carbohydrate metabolism, amino acid metabolism, energy metabolism, translation, membrane transport, metabolism of cofactors and vitamins, replication and repair, nucleotide metabolism, signal transduction (Figure 6A). The analysis found the metabolism of cofactors and vitamins pathway to be elevated with increasing essential oil addition and enriched in HIG (*p* < 0.05) (Figure 6B).

### 3.8. Correlation Analysis of Rumen Microorganisms with Growth Performance and Serum Immunoglobulin in Calves

Spearman correlation coefficients were used to correlate calf average daily gain and three immunoglobulin concentrations with rumen microflora, and the results showed that *unclassified__c__Clostridia* and *Shuttleworthia* were positively correlated with calf ADG (*p* < 0.05), *norank__f__Oscillospiraceae* were negatively correlated with calf ADG (*p* < 0.05), and *Ruminococcus_gauvreauii_group* was negatively correlated with calf ADG (*p* < 0.01). *Christensenellaceae_R-7_group* was positively correlated with calf serum IgA concentration (*p* < 0.01), and *Christensenellaceae_R-7_group* was positively correlated with calf serum IgA concentration (*p* < 0.05). *Shuttleworthia* was positively correlated with calf serum IgA concentration (*p* < 0.01). *Coprococcus*, *Ruminococcus_gauvreauii_group*, and *Olsenella* were correlated with calf serum IgA concentration (*p* < 0.05), and *norank_f__Eubacterium_coprostanoligenes_group* showed a negative correlation with calf serum IgA concentration (*p* < 0.01). *Syntrophococcusp* showed a correlation with calf serum IgG concentration (*p* < 0.05); *unclassified__c__Clostridia* and *Shuttleworthia* were positively correlated with calf serum IgM concentrations (*p* < 0.05), and *Ruminococcus_gauvreauii_group* was negatively correlated with calf serum IgM concentrations (*p* < 0.01) (Figure 7).

### 3.9. Core Species Screening for Rumen Microbiota in Calves

Spearman’s correlation coefficient was used with the absolute value of correlation coefficient ≥ 0.5, *p* < 0.05, and species-to-species correlation analysis was performed at the genus level. The results showed that (Figure 8) there were 47 species in the CON, of which the core species was *g_Mogibacterium* (degree = 7), and there existed positive correlations of 40 and negative correlations of 23. The OEO treatment group had 44 species, of which the core species was *g_Ruminococcus* (degree = 8), with 51 positive and 22 negative correlations.

## 4. Discussion

### 4.1. Effect of Different Oregano Essential Oil Additions on Calf Growth Performance

Growth performance is closely related to animal health, and improving the growth performance of animals is a critical factor in promoting their healthy growth [26]. The addition of oregano essential oil to the calf diet significantly enhanced the open feed intake and daily weight gain of calves [27]. The addition of a mixture of cinnamon, thyme and peppermint essential oils fed in the basal diet of goats significantly increased the weaning weight and total body weight of goats as compared to the control group [28]. When studying dairy calves, it was found that supplementation of the diet with essential oils significantly increased calf weight gain [29]. The results of the above studies are consistent with the results of the present study. The results of a number of studies have shown that oregano essential oil can increase feed intake in livestock [30] and can improve rumen digestibility by modulating the development of epithelial tissues and the microbial composition of cattle [25]. This process promotes the calf’s feeding frequency as well as the improvement of the digestive system, so the calf end weight and daily weight gain were significantly higher than in the CON group. Thus, calves can access a higher level of nutrition and their growth and development are promoted.

### 4.2. Effect of Different Oregano Essential Oil Additions on Calf Serum Immunoglobulin

Immunoproteins are an indicator of the immune ability of the animal organism, and they are also essential substances for the animal organism to resist the attack of pathogens; they improve the organism’s immunity [31]. IgA is synthesized by gastrointestinal lymphocytes and is classified into serotypes and secretory types [32], which are the primary antibodies of the mucosal immune mechanism and are involved in the local immune response of the animal organism [33]. Serum IgG levels are critical for calves, with increased serum IgG concentrations reducing the risk of calf morbidity and mortality [34] and their concentration significantly affecting late calf weight gain [35]. IgM is secreted by the B cells of the spleen and is divided into membrane-bound and secreted types mainly found in animals’ blood [36]. IgM recognizes and destroys bacteria in the bloodstream and is the first to undergo an immune response when exposed to an antigen, characterized by a short half-life, early appearance, and rapid disappearance [37]. In the present study, there was no difference in immunoglobulin concentrations among all groups of calves at the beginning of the trial. At the end of the trial, the serum concentrations of all three immunoglobulins were higher in the treated calves than in the control group, and there was a significant trend in the serum concentrations of IgA and IgM in the treated calves. Studies have shown that dietary nutrients play an essential role in organismal immunity. Animals can obtain nutrients related to the growth and immunomodulation of the organism, such as proteins, carbohydrates, fatty acids, vitamins, minerals, etc., through food intake [38]. These nutrients interact with lymphoid tissues in the gut and play a role in immunity and inflammation by modulating the expression of Toll-like receptors, pro-inflammatory cytokines, and anti-inflammatory cytokines, thereby interfering with immune cell crosstalk and signaling [39]. This means that the organism receives a higher level of nutrition with a correspondingly better immune system. As mentioned in the previous section, oregano essential oil can improve feeding digestion and absorption by promoting the development of epithelial tissues to enable calves to obtain higher levels of nutrients. With adequate protein intake, energy, and micronutrients, these nutrients initiate and regulate the adaptive immune response by modulating antibody levels [40], resulting in higher disease resistance.

### 4.3. Effect of Oregano Essential Oil Supplementation on Rumen Microbial Community in Calves

Microbial community diversity was interpreted using the Alpha diversity index (Chao1; Shannon; Simpson; PD and coverage). The results of this study showed that supplementation with oregano essential oil significantly affected the abundance and diversity of rumen microorganisms in calves, according to the results of the Chao1, Shannon and Simpson indices; the abundance and diversity of rumen microorganisms in calves increased with the increase in the addition amount, the highest species abundance was found in the HIG, and the levels of species diversity in the MID and the HIG were close to one another, which indicated that the addition amount of 6 g and above would significantly affect the rumen microbial community of calves. The similarity of microbial community composition between groups was interpreted using a principal coordinate analysis (PCoA), which showed that different oregano essential oil additions did not significantly affect species composition. At the phylum and genus levels, the dominant flora species were similar between groups with various additions. The relative abundance of Olsenella in the HIG was significantly lower, and that of Shuttleworthia was significantly higher than in the other groups. In this study, we found that the core species composition of calf rumen microorganisms does not change after oregano essential oil supplementation, the species composition is relatively stable, and the primary way essential oils affect calf rumen microorganisms is through regulating the relative abundance of some species.

### 4.4. Differences in Rumen Microbial Composition at Different Taxonomic Levels

According to the LEfSe screen, the biomarker that plays a crucial role in CON is Erysipelatoclostridiaceae, which is associated with colonic diseases and is a potential pathogen. The abundance of this species is significantly increased in individuals suffering from functional bowel disorders, and oregano can alleviate this condition by remodeling the microflora of the organism and modulating the metabolism of bile acids and short-chain fatty acids in this disease [41]. In this study, the highest abundance of Erysipelatoclostridiaceae was found in the CON group, and the lowest abundance was found in the MID group, suggesting that oregano essential oil can reduce the risk of intestinal diseases and the most significant mitigating effect of HIG by modulating the structure of rumen microflora in calves. The treatment group supplemented with oregano essential oil had more biomarkers that played a key role, among them bacteria of the genus Turicibacter, a critical member of the mammalian gut microbiota that is correlated with changes in dietary fat and body weight [42] and which was also positively correlated with apparent digestibility [43]. It has been shown that metabolite plasma levels can be regulated by increasing the abundance of Turicibacte to improve metabolic disorders [44]. Romboutsia can regulate glucose metabolism through the production of short-chain fatty acids to stimulate insulin secretion, and it can also increase the carbohydrate content of feeds, which can have a lowering effect on blood glucose [45]. Peptostreptococcaceae is positively correlated with intestinal GSH-Px activity [46], and its reduced abundance is favorable to the body’s immunity [47]. The elevated abundance of Clostridium_sensu_stricto_1 leads to increased diarrhea in calves [48]. Clostridiales are associated with animal organismal immunity and digestion [49]. Coriobacteriale and Coriobacteriia can increase gut microbial metabolite SCFAs [50]. Atopobiaceae are important in systemic inflammation and metabolic disorders [51]. g_Defluviitaleaceae_UCG-011 was detected at a higher abundance in the microbial composition of non-diarrheic calves [52]. Olsenella abundance and enhancement can promote and elevate the levels of the body’s anti-inflammatory cytokine IL-10 [53]. Shuttleworthia is considered to be a unique bacterium for calves after weaning, and the genus Shuttleworthia is highly positively correlated with MCP, propionate, total volatile fatty acids (TVFA), glucosidase, acetate, and butyrate [54]. Hungateiclostridiaceae acts as an acid-producing bacterium that enhances the production and accumulation of VFA in the body [55]. Saccharofermentan is an anaerobic Gram-negative bacterium that converts glucose to acetic acid [56]. Streptococcus was significantly and positively correlated with growth rate and sugar and protein metabolites such as propionic acid, butyric acid, maltoketone sugars, and amino acids in young ruminants [57]. Oscillospirales were reported to be a growth-beneficial species [58], and feeding with oregano essential oil significantly increased the abundance of this bacterium [59]. This experiment showed that the bacteria that changed in relative abundance after oregano essential oil treatment were mainly concentrated in acid-producing bacteria, which could increase the production of VFA in the gastrointestinal tract. VFA is an important signaling molecule in the rumen microbiota that can maintain a steady state between the host and microbial colonies by increasing the diversity of microbial colonies and maintaining the integrity of the rumen epithelium in healthy animals while enhancing the activity of the body’s immune barrier [60]. Microbially fermented short-chain fatty acids also have anti-inflammatory properties and may protect intestinal health [61]. At the same time, supplemental essential oils down-regulated the abundance of several species that are pathogenic to the organism (Erysipelatoclostridiaceae, Peptostreptococcaceae, Clostridium_sensu_stricto_1). This process allows the animal organism to better utilize nutrients and obtain higher levels of VFA under the mutual coordination and action of microorganisms, thus achieving the purpose of promoting the establishment and improvement of calf gastrointestinal function, maintaining gastrointestinal homeostasis, and enhancing the intestinal barrier function, which is in line with the results reported by some previous studies [25,62,63,64].

### 4.5. Modulation of Rumen Microflora in Calves by Oregano Essential Oil Supplementation

This study found that providing oregano essential oil to calves significantly affected their rumen microbiota. According to the results of microbial function prediction, oregano essential oil could promote the metabolic utilization of amino acids and vitamins in calves and increase the body’s resistance to diseases, and this change increased with the addition of the essential oil. Vitamins are necessary for the organism, playing a crucial role in growth and development, immunocompetence, and lipid metabolism [65]. It has been reported that the nutrition provided by vitamins promotes growth and immune function in developing calves [66] and that vitamins are responsible for the development of the organism, immunity, and the maintenance, regulation, and repair of tissues through their various metabolites [67]. In addition, vitamins are cofactors for different critical metabolic reactions in organisms [68].

Through correlation analysis, *unclassified_c_Clostridia* and *Shuttleworthia Christensenellaceae_R-7_group* were screened for positive correlation with ADG and immunoglobulin, which contributed to calf growth performance and immunocompetence. In previous reports, *Clostridia* was shown to be associated with the degradation of cellulose in the rumen, which can be achieved through the synergistic action of microorganisms and enzymes [69] and the fermentation of cellulose into other nutrients [70], *Shuttleworthia* mainly serves to provide the organism with a more excellent supply of short-chain fatty acids, as mentioned previously. *Christensenellaceae _R-7_group* is associated with ruminant health and digestion, and its role in butyric acid production [71] and the promotion of rumen development was mentioned in a related study [72].

Through the screening of the correlation network diagram between species, the core species with the highest number of inter-species associations in CON was *g_Mogibacterium*, a pro-inflammatory bacterium present in the digestive tract of ruminants [73], and in the group supplemented with oregano essential oil, the core species was *g_Ruminococcus*, which plays a significant role in the rumen of ruminants in the production of short-chain fatty acids [74] and the degradation of cellulose [75]. This suggests that oregano essential oil modulates the structural function of rumen microbes in calves and promotes gastrointestinal digestion.

## 5. Conclusions

As a natural plant additive supplemented to calves, oregano essential oil promotes calf growth and development and positively affects serum antibody levels. Analysis of the rumen microbial community revealed that oregano oil could significantly increase the diversity of rumen microorganisms and regulate the abundance of some microbial species, which could play a positive role in nutrient digestion and metabolism, the production of short-chain fatty acids, and the enhancement of gastrointestinal immunity. Under the conditions of this experiment, supplementation with 8 g of oregano essential oil per calf per day was the most beneficial for regulating rumen microorganisms.

## Figures and Tables

**Figure 1 animals-14-00820-f001:**
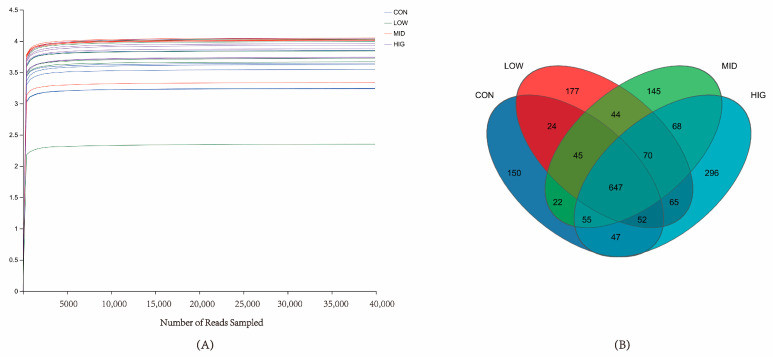
Sequencing base results. (**A**) Species dilution curves: horizontal coordinates represent the amount of randomly selected sequencing data; vertical coordinates are diversity indices; (**B**) Species Venn diagrams.

**Figure 2 animals-14-00820-f002:**
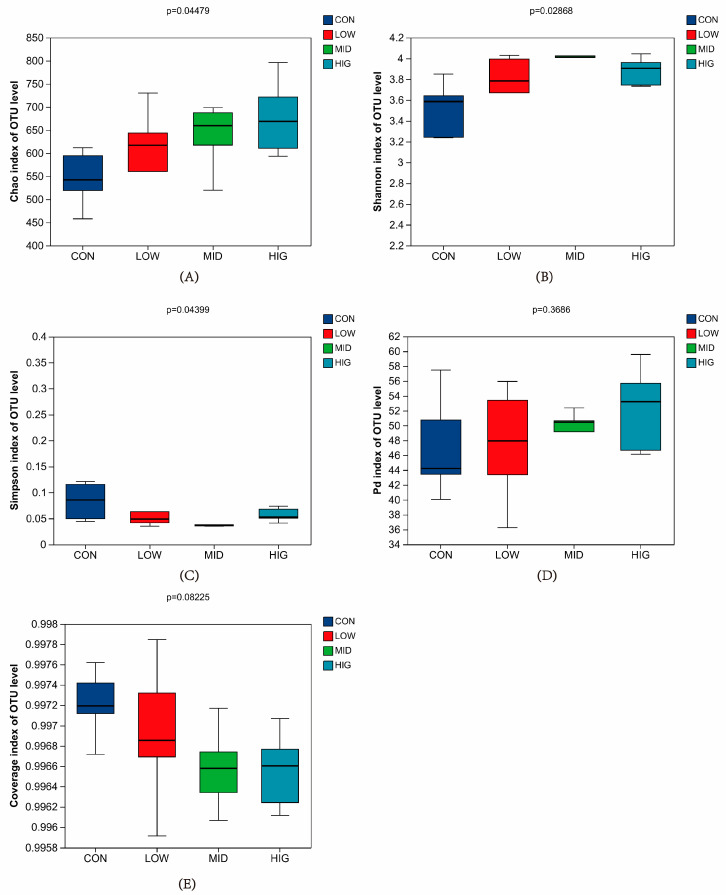
Alpha diversity box chart. (**A**) Chao1 index of colony species abundance; (**B**) Shannon index of colony species diversity; (**C**) Simpson index of colony species diversity (**D**); PD index of colony species genealogy diversity; (**E**) Good’s species coverage index.

**Figure 3 animals-14-00820-f003:**
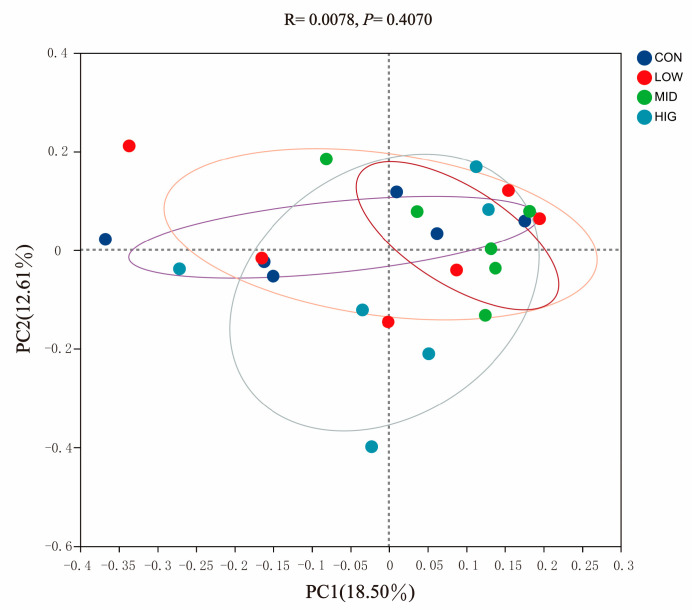
Principal coordinate analysis (PCoA).

**Figure 4 animals-14-00820-f004:**
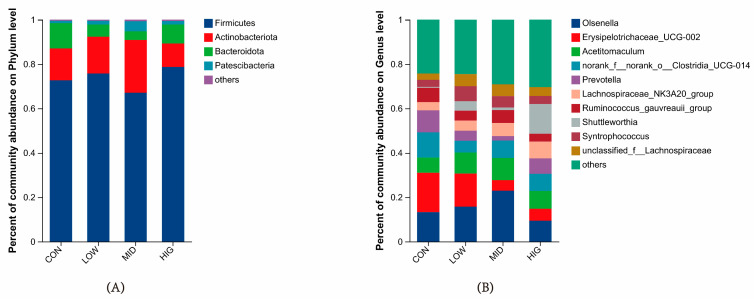
Microbial composition of the calf gut. (**A**) Bar chart of species composition at the phylum level, with the horizontal coordinates being the groupings and the vertical coordinates being the percentage of species abundance; (**B**) Bar chart of species composition at the genus level, with the same groupings and the vertical coordinates being the percentage of species abundance.

**Figure 5 animals-14-00820-f005:**
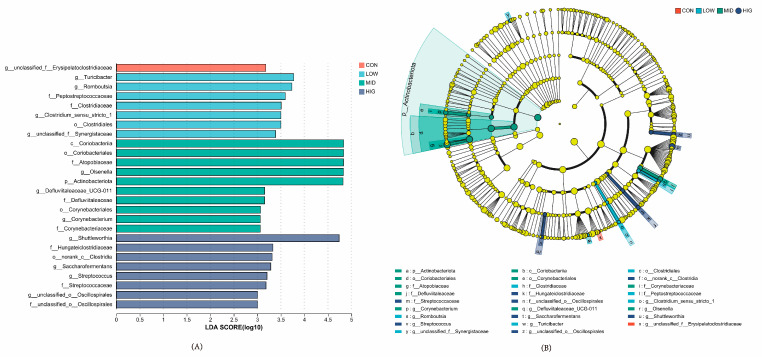
Species differences among calf gut microbial groups. (**A**) Developmental dendrogram; different color nodes indicate microbial taxa that are significantly enriched in the corresponding groups and have significant effects on intergroup differences. (**B**) LDA discrimination histogram.

**Figure 6 animals-14-00820-f006:**
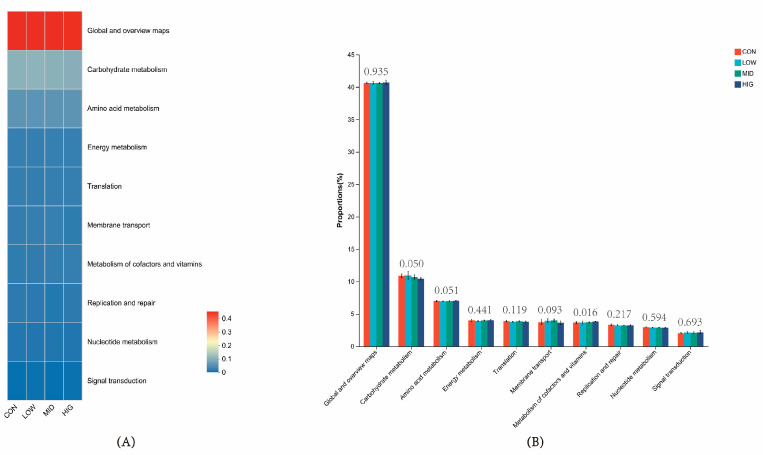
Predicted function of rumen microbial PICRUSt2 in calves. (**A**) Functional pathways on KEGG Pathway 2 (Top10) (**B**) Differential functional pathways between groups.

**Figure 7 animals-14-00820-f007:**
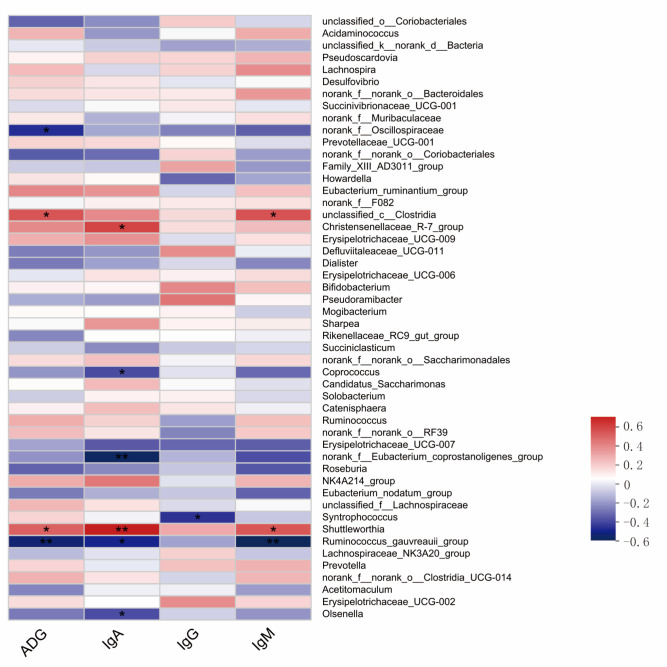
Correlation heatmap. Correlation of rumen microorganisms with average daily gain (ADG) and serum concentrations of IgG, IgM and IgA in calves; * represents *p* ≤ 0.05, ** represents *p* ≤ 0.01.

**Figure 8 animals-14-00820-f008:**
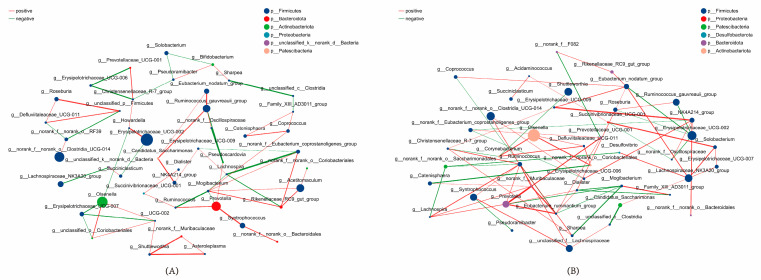
Correlation network diagram. (**A**) control species correlation; (**B**) oregano essential oil (OEO) treatment group species correlation.

**Table 1 animals-14-00820-t001:** Ingredient composition and nutrient levels of the basal diets (dry matter basis) %.

Ingredient Composition	Content	Nutrient Level	Content
Corn	37.57	DM ^1^	90.87
Soybean meal	27.21	CP ^2^	21.58
Bran	12.40	CF ^3^	4.18
Cottonseed meal	6.80	Ash	5.60
Puffed soybeans	2.50	NDF ^4^	22.58
Whey powder	4.00	ADF ^5^	8.86
Molasses	4.00	Ca ^6^	1.08
Calcium carbonate	2.20	P ^7^	0.48
Soybean oil	0.80		
Nacl	0.80		
Calcium biphosphate	0.90		
Magnesium oxide	0.10		
Yeast selenium	0.02		
Premix	0.70		
Total	100		

^1^ = Dry matter, ^2^ = Crude protein, ^3^ = Crude fat, ^4^ = Neutral detergent fiber, ^5^ = Acid detergent fiber, ^6^ = Calcium, ^7^ = Phosphorus.

**Table 2 animals-14-00820-t002:** Effect of oregano essential oil on calf growth performance (kg).

Index	Treatment				SEM	*p*-Value
	CON	LOW	MID	HIG		
Initial weight	92.42	91.17	91.50	92.58	0.47	0.689
Final weight	153.92 c	170.92 b	168.08 bc	177.42 a	3.15	0.046
Average daily gain	1.10 c	1.42 b	1.37 bc	1.51 a	0.05	0.037

a, b, c Means in the same row with unlike superscripts differ, *p* < 0.05.

**Table 3 animals-14-00820-t003:** Effect of oregano essential oil on calf serum immunoglobulin (g/L).

Index	Days	Treatment				SEM	*p*-Value
		CON	LOW	MID	HIG		
IgA	0	1.28	1.25	1.32	1.29	0.03	0.937
	56	1.99 b	2.12 ab	2.09 ab	2.28 a	0.04	0.028
IgG	0	8.16	7.63	7.23	7.59	0.20	0.475
	56	13.44	13.27	14.38	14.15	0.30	0.509
IgM	0	1.10	0.96	1.02	0.96	0.04	0.550
	56	1.65 b	1.83 ab	1.92 a	1.99 a	0.47	0.045

a, b Means in the same row with unlike superscripts differ, *p* < 0.05.

**Table 4 animals-14-00820-t004:** Species composition of rumen microorganisms at different levels in calves.

Index	Relative Abundance (%)				
	CON	LOW	MID	HIG	*p*-Value
Phylum level					
Firmicutes	72.68 ± 15.85	75.76 ± 6.55	67.05 ± 10.99	78.66 ± 10.63	0.311
Actinobacteriota	14.33 ± 10.80	16.56 ± 7.71	23.83 ± 6.29	10.6 ± 5.95	0.042
Bacteroidota	11.47 ± 12.48	5.47 ± 6.35	3.87 ± 1.67	8.61 ± 9.58	0.397
Patescibacteria	0.99 ± 0.61	1.70 ± 0.44	4.58 ± 7.64	1.54 ± 0.64	0.236
Proteobacteria	0.23 ± 0.24	0.17 ± 0.17	0.18 ± 0.07	0.20 ± 0.15	0.928
Genus level					
*Olsenella*	13.28 ± 10.23	15.73 ± 7.49	22.9 ± 6.08	9.46 ± 5.97	0.039
*Erysipelotrichaceae_UCG-002*	17.74 ± 13.49	14.96 ± 22.71	4.80 ± 3.64	5.34 ± 5.20	0.354
*Acetitomaculum*	6.87 ± 3.72	9.46 ± 5.37	10.04 ± 2.81	8.00 ± 4.48	0.646
*norank_f_norank_o_Clostridia_UCG-014*	11.39 ± 9.35	5.28 ± 2.87	7.77 ± 5.24	7.77 ± 3.52	0.435
*Prevotella*	9.88 ± 12.52	4.56 ± 6.03	2.0 ± 1.23	6.97 ± 9.65	0.717
*Lachnospiraceae_NK3A20_group*	3.67 ± 1.42	4.58 ± 2.54	5.95 ± 1.76	7.54 ± 7.09	0.286
*Ruminococcus_gauvreauii_group*	6.51 ± 3.71	4.44 ± 2.59	5.84 ± 1.84	3.57 ± 0.74	0.227
*Shuttleworthia*	0.39 ± 0.31	4.30 ± 7.10	1.15 ± 0.90	13.38 ± 10.55	0.003
*Syntrophococcus*	3.17 ± 1.96	6.76 ± 3.86	5.12 ± 2.09	3.58 ± 2.15	0.257
*unclassified_f_Lachnospiraceae*	2.85 ± 1.68	5.42 ± 5.99	5.31 ± 3.06	4.03 ± 2.82	0.528

## Data Availability

Data from this study are included in the article; please contact the corresponding author if you need more information.

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
