# Peer review of "Oregano Essential Oil as a Natural Plant Additive Affects Growth Performance and Serum Antibody Levels by Regulating the Rumen Microbiota of Calves"

_animals, 2024, doi:10.3390/ani14060820_

Round 1
Reviewer 1 Report
Comments and Suggestions for Authors
The manuscript authored by Luo et al. reported the effect of oregano essential oil (OEO) on growth, antibody titer, and microbiota in calves. Overall, the quality of the manuscript is now enough to be accepted in the journal ‘Animals’, although the microbiota analysis seems fine. The authors collected rumen fluid and analyzed the microbial community at 56 days after the feeding. The individual calf has unique microbiota at birth and develops them over time, and it depends on whose they fed colostrum and cow’s condition. To better elucidate the effect of OEO and changes in microbiota by supplementation of OEO, the authors should have analyzed the community before feeding them as well. Throughout the manuscript, 1) the design of the experiment seems poor, although the analysis seems okay, 2) more discussion on specific microorganisms that showed significant changes at 56 days since it is not clear how each of them could affect the stability or health of the calves. Lastly, please check if the manuscript needs an IACUC #.
Comments on the Quality of English LanguageMinor editing of English language required
Author Response
Thank you for taking your valuable time to review our article and for recognizing our work. We have rewritten and rechecked the article based on the suggestions you made. Through your review report, we also found the shortcomings of our work, which also provided us with better ideas and methods for our future research work. Thank you again for your work. We have tried our best to reply to your questions, and we have marked all the corrections in the article in red, so we hope you can review them.
The reply is as follows:
1. To better elucidate the effect of OEO and changes in microbiota by supplementation of OEO, the authors should have analyzed the community before feeding them as well.
Reply: Thank you very much for your comments, which we failed to take into account when designing the test. We have given your comments some serious thought after receiving them. As you suggested, microbiological analyses of calves before feeding would make our trial data more representative. We have also referred to this line of thinking when designing our trial, as we referred to some literature [1,2,3], which also did not sample and analyze pre-treatment animals when conducting the relevant studies. Now the test has been over for a long time, the batch of test animals has been unable to test, and there are large individual differences between individual animals, can not guarantee the same conditions, this part of the test can not go to add. Unfortunately, this line of thought was not considered. In this trial, we found that the significant effect of oregano essential oil was to increase VFA production. We are already working on a follow-up study that will combine microbes with metabolites and we will add your comments to this study by sampling and analyzing calves prior to treatment. Sincerely, thank you for this question, it is extremely helpful and provides a valuable approach to our work. Thank you again for reviewing our article and pointing out this issue!
Reference:
[1] Schären M, Drong C, Kiri K, Riede S, Gardener M, Meyer U, Hummel J, Urich T, Breves G, Dänicke S. Differential effects of monensin and a blend of essential oils on rumen microbiota composition of transition dairy cows. J Dairy Sci. 2017 Apr;100(4):2765-2783. doi: 10.3168/jds.2016-11994. Epub 2017 Feb 1. PMID: 28161182.
[2] Zhang R, Wu J, Lei Y, Bai Y, Jia L, Li Z, Liu T, Xu Y, Sun J, Wang Y, Zhang K, Lei Z. Oregano Essential Oils Promote Rumen Digestive Ability by Modulating Epithelial Development and Microbiota Composition in Beef Cattle. Front Nutr. 2021 Nov 9;8:722557. doi: 10.3389/fnut.2021.722557. PMID: 34859026; PMCID: PMC8631176.
[3] Hales KE. Relationships between digestible energy and metabolizable energy in current feedlot diets. Transl Anim Sci. 2019 Jun 25;3(3):945-952. doi: 10.1093/tas/txz073. PMID: 32704858; PMCID: PMC7200974.
2. more discussion on specific microorganisms that showed significant changes at 56 days since it is not clear how each of them could affect the stability or health of the calves.
Reply: Thank you for your comments. Based on your comments, we have revised and added to the discussion in the Differential Species section. (L423-430,L447-468)
3. Lastly, please check if the manuscript needs an IACUC #.
Reply: Thank you for your question. We have provided the relevant materials to the editor.
Reviewer 2 Report
Comments and Suggestions for Authors
The article by Luo et al., titled "Oregano essential oil as a natural plant additive improves calf growth and health by regulating the rumen microbiota," discusses the topic of natural feed additives. The research covered three aspects: the impact on calf growth performance and rumen microbiota, as well as antibody concentration studies. The work is interesting, and the results are presented clearly. However, it is not without certain shortcomings.
The title of the article does not reflect the scope of the conducted research. The study did not focus on improving calf health; in this regard, only antibody concentrations were examined (increased antibody levels also occur after infection). The obtained results may not necessarily be due to the effect of the oil on rumen microbiota. The action could have a direct effect on antibody production and metabolic pathways. Considering the above, the authors should revise and clarify the title.
Furthermore, there were also some errors or lack of precise descriptions:
Please clarify what "free from diseases" means. Were studies conducted in this area?
What does "rumen catheter" mean? Did the authors implant permanent rumen catheters in the calves?
There is no information about the gender of the animals or their distribution in the groups. Gender in cattle can affect production indicators (calf growth performance).
Oregano oil was not subjected to quantitative and qualitative analysis. Plant oils can vary in composition, and consequently, in their effects.
Author Response
Thank you for reviewing our article and pointing out the deficiencies in our work, which is crucial for us to improve the quality of our article and conduct future research. We have responded to and revised the questions you raised, and the changes and additions are marked in red in the text in the hope that they will be recognised by you, and thank you once again for your work for us.
The reply is as follows:
1.The title of the article does not reflect the scope of the conducted research. The study did not focus on improving calf health; in this regard, only antibody concentrations were examined (increased antibody levels also occur after infection). The obtained results may not necessarily be due to the effect of the oil on rumen microbiota. The action could have a direct effect on antibody production and metabolic pathways. Considering the above, the authors should revise and clarify the title.
Reply: Thank you very much for your comments, which have helped us immensely in refining the manuscript. As you said, the level of antibodies does not fully reflect the health status of the calves, and we comment on your suggestion to change the title of the paper to: Oregano essential oil as a natural plant additive affects growth performance and serum antibody levels by regulating the rumen microbiota of calves. And revise the description of immunocompetence in the text to antibody levels. (L112,L507)
2.Please clarify what "free from diseases" means. Were studies conducted in this area?
Reply: Thank you for pointing this out, disease free in this case means that we excluded individuals suffering from respiratory diseases (colds, pneumonia), gastrointestinal diseases (dyspepsia, diarrhoea) and hoof and limb diseases when selecting calves for the trial, and calves entered into the trial group were healthy and not suffering from any of these diseases, the descriptions were not specific enough in the article, and have been modified in the article. (L119-121)
3.What does "rumen catheter" mean? Did the authors implant permanent rumen catheters in the calves?
Reply: Thank you for your question, we did not implant a permanent catheter in the calf in this trial, we used the method of inserting a rumen catheter. This is done by inserting the catheter into the calf's mouth and extending it into the rumen to extract rumen fluid as the calf swallows. This method causes relatively minor injury and stress to the calf. We reviewed the literature (DOI: 10.3389/fvets.2022.875741) and the method of collecting rumen fluid in the paper was similar to ours. The description of the rumen fluid collection method has been modified in the article with reference to the description in the literature (L151).
4.There is no information about the gender of the animals or their distribution in the groups. Gender in cattle can affect production indicators (calf growth performance).
Reply: Thank you for pointing out the problem, we uniformly chose male calves for the study in that trial and have added a description of gender to the article, thank you for your comments. (L118)
5.Oregano oil was not subjected to quantitative and qualitative analysis. Plant oils can vary in composition, and consequently, in their effects.
Reply:Your suggestion is very interesting, we have been studying the application of oregano essential oil in livestock and food, the oil has been previously analysed qualitatively and quantitatively in other studies using GC/MS methods, we have added the main components of oregano essential oil in the manuscript based on your comments, thank you again for your suggestion! (L125-126)
Reviewer 3 Report
Comments and Suggestions for Authors
Lines 68-69 check tipos
Line 121. Cow??
Line 131. I would suggest adding the age of animals in brackets after day of trial because it is confusing: real age vs day of study. When does the study finish? 56 days of supplementation?
Line 138. Please, explain how the feed was offered How oregano was included in the feed?
Line 162. As you made so many comparisons, do you consider repeating the analysis using false detection rate? This is the critical point, because it would probably modify the results and discussion
Line 330. Was there any issue with the oregano flavour?
Comments on the Quality of English LanguageLines 68-69 check tipos
Author Response
Thank you very much for taking the time to review our article. We apologize for our carelessness in writing and we have revised the text for grammatical errors. Based on your comments, we have revised the problems in the text, and the revisions are marked in red in the text. We hope you can review our revisions and thank you again for your time.
The reply is as follows:
1. Lines 68-69 check tipos
Reply: Thank you very much for identifying this problem and pointing it out to us, I apologize for our carelessness, and we have revised the formulation of this sentence. (L69-70)
2. Line 121. Cow??
Reply: Thank you for raising the issue and again we apologize for our carelessness, have amended cows to calves in the article. (L130)
3. Line 131. I would suggest adding the age of animals in brackets after day of trial because it is confusing: real age vs day of study. When does the study finish? 56 days of supplementation?
Reply: We have taken your comments into account by describing the trial date and calf age in more detail (L128-130), adding the calf's age in the paragraph describing the study day. Thank you for your suggestions!
4. Line 138. Please, explain how the feed was offered How oregano was included in the feed?
Reply: Greetings! As adding oregano to the feed under the conditions of our trial could not accurately guarantee the calf's intake of the essential oil, we chose to infuse the calf with oregano essential oil in the course of our trial, which we have described in more detail in the text, thank you once again for your valuable comments. (L134-135)
5. Line 162. As you made so many comparisons, do you consider repeating the analysis using false detection rate? This is the critical point, because it would probably modify the results and discussion.
Reply: Thank you for pointing out this problem. In the description of the analytical methods section, we described in detail the software versions and calculation methods used for each type of analysis, which did not describe the FDR test. In fact, when analyzing microbial diversity and species differences between groups, we used the FDR to correct for multiple testing of P values. In response to your question, we have added information on the correction for multiple testing in the Methods section of the paper, and thank you again for identifying and pointing out this problem. (L194-196)
6.Line 330. Was there any issue with the oregano flavour?
Reply: Thank you for pointing this out, as oregano essential oil has an aromatic odour, we speculated that this odour could stimulate calves' appetite and increase feed intake. Based on the questions you raised, we found no basis for this speculation. We have reviewed some of the references again and have revisited this section. (L354-357)
Round 2
Reviewer 3 Report
Comments and Suggestions for Authors
Calves in the treatment group were dosed with oregano essential oil at the end of 134 the daily morning feed to ensure correct intake.
Regarding the dosing, please, explain what did you used to infuse the dose, it is not clear the "how" in your sentence
Author Response
Thank you to the reviewers for their work and dedication, and we apologise for some very basic errors. We have made the appropriate changes in the article based on your suggestions. We look forward to your review. Thank you again for your work on our article and we hope you have a good life and work!
Comments and Suggestions for Authors: Calves in the treatment group were dosed with oregano essential oil at the end of 134 the daily morning feed to ensure correct intake. Regarding the dosing, please, explain what did you used to infuse the dose, it is not clear the "how" in your sentence
Reply: Thank you to the reviewers for their comments. We checked the presentation here. We apologise for not describing the method of supplementation of essential oils accurately enough as our native language is not English. We are gavaging oregano essential oil directly to calves. We have reviewed some literature and the word "gavaged" is the correct description. We have made changes in the article. (L126-127)